# HER2-CD3-Fc Bispecific Antibody-Encoding mRNA Delivered by Lipid Nanoparticles Suppresses HER2-Positive Tumor Growth

**DOI:** 10.3390/vaccines12070808

**Published:** 2024-07-21

**Authors:** Liang Hu, Shiming Zhang, John Sienkiewicz, Hua Zhou, Robert Berahovich, Jinying Sun, Michael Li, Adrian Ocampo, Xianghong Liu, Yanwei Huang, Hizkia Harto, Shirley Xu, Vita Golubovskaya, Lijun Wu

**Affiliations:** 1Promab Biotechnologies, 2600 Hilltop Drive, Richmond, CA 94806, USA; liang.hu@promab.com (L.H.); alan.zhang@promab.com (S.Z.); john.sienkiewicz@promab.com (J.S.); hua.zhou@promab.com (H.Z.); robert.berahovich@promab.com (R.B.); sunnie.sun@promab.com (J.S.); michael.li@promab.com (M.L.); adrian.ocampo@promab.com (A.O.); xianghong.liu@promab.com (X.L.); yanwei.huang@promab.com (Y.H.); hizkia.harto@promab.com (H.H.); shirley.xu@promab.com (S.X.); 2Forevertek Biotechnology, Janshan Road, Changsha Hi-Tech Industrial Development Zone, Changsha 410205, China

**Keywords:** HER2, cancer, bispecific antibody, CD3, lipid nanoparticle, immunotherapy

## Abstract

The human epidermal growth factor receptor 2 (HER2) is a transmembrane tyrosine kinase receptor and tumor-associated antigen abnormally expressed in various types of cancer, including breast, ovarian, and gastric cancer. HER2 overexpression is highly correlated with increased tumor aggressiveness, poorer prognosis, and shorter overall survival. Consequently, multiple HER2-targeted therapies have been developed and approved; however, only a subset of patients benefit from these treatments, and relapses are common. More potent and durable HER2-targeted therapies are desperately needed for patients with HER2-positive cancers. In this study, we developed a lipid nanoparticle (LNP)-based therapy formulated with mRNA encoding a novel HER2-CD3-Fc bispecific antibody (bsAb) for HER2-positive cancers. The LNPs efficiently transfected various types of cells, such as HEK293S, SKOV-3, and A1847, leading to robust and sustained secretion of the HER2-CD3-Fc bsAb with high binding affinity to both HER2 and CD3. The bsAb induced potent T-cell-directed cytotoxicity, along with secretion of IFN-λ, TNF-α, and granzyme B, against various types of HER2-positive tumor cells in vitro, including A549, NCI-H460, SKOV-3, A1847, SKBR3, and MDA-MB-231. The bsAb-mediated antitumor effect is highly specific and strictly dependent on its binding to HER2, as evidenced by the gained resistance of A549 and A1847 *her2* knockout cells and the acquired sensitivity of mouse 4T1 cells overexpressing the human HER2 extracellular domain (ECD) or epitope-containing subdomain IV to the bsAb-induced T cell cytotoxicity. The bsAb also relies on its binding to CD3 for T-cell recruitment, as ablation of CD3 binding abolished the bsAb’s ability to elicit antitumor activity. Importantly, intratumoral injection of the HER2-CD3-Fc mRNA-LNPs triggers a strong antitumor response and completely blocks HER2-positive tumor growth in a mouse xenograft model of human ovarian cancer. These results indicate that the novel HER2-CD3-Fc mRNA-LNP-based therapy has the potential to effectively treat HER2-positive cancer.

## 1. Introduction

The human epidermal growth factor receptor 2 (HER2/neu, also called ErbB2) is a member of the HER family of tyrosine kinases, along with HER1 (also called epidermal growth factor receptor (EGFR), ErbB1), HER3 (ErbB3), and HER4 (ErbB4) [1]. The HER family members are structurally related membrane glycoproteins that consist of an extracellular cysteine-rich domain with ligand binding sites, a short single transmembrane region, and an intracellular domain with tyrosine kinase catalytic activity [2]. Upon ligand binding, the HER proteins undergo conformational changes, triggering homo- and heterodimerization such as HER2-HER2, HER2-HER3, and HER2-HER4 [3]. Dimerization is essential for the activation of intracellular tyrosine kinase catalytic activity, leading to autophosphorylation of tyrosine residues in the C-terminal tail [4]. The activated HER proteins further phosphorylate a variety of substrates to activate numerous signaling pathways, including mitogen-activated protein kinase (MAPK), phosphoinositide-3-kinase (PI3K)/Akt, and phospholipase C-γ (PLC-γ), resulting in alterations in cell proliferation, invasion, and survival [5,6]. Among the HER family members, HER2 exhibits the most potent kinase activity, thereby driving robust downstream signaling [7,8]. HER2 does not directly bind to any known ligand but can be activated through dimerization with other HER family members by many ligands, such as epithelial-growth-factor-like ligands and neuregulins [9,10]. 

HER2 is widely expressed in various non-hematopoietic tissues and is essential for the development of many organs and tissues, including the nervous system, mammary tissue, lung, gastrointestinal tract, and heart [11,12,13,14,15]. However, many preclinical and clinical studies have identified abnormal HER2 protein expression, gene amplification, and mutations in various malignancies of epithelial origin, especially cancers of the breast, stomach, ovary, and bladder [16,17,18]. HER2 is overexpressed in 15–30% of invasive breast cancers [19], 10–30% of gastric cancers [20], and 20–30% of ovarian cancers [21]. Some breast tumors have been found to contain cell subpopulations with more than 100 copies of the *HER2* gene, resulting in 100-fold elevated HER2 protein levels [22]. Higher expression and gene amplification of HER2 are closely associated with increased tumor aggressiveness, worse prognosis, and shorter overall survival [23,24]. *HER2* gene mutations, predominantly missense substitutions and insertions, are reported to occur at a frequency of 3.5% across approximately 400 types of cancer, and usually result in HER2 protein activation and cellular survival signaling [25]. Consistent with these clinical findings, the overexpression of HER2 in different cell lines and animal models induces malignant transformation and tumorigenesis [26,27,28,29]. Therefore, HER2 expression is an important biomarker of predicative implication and therapeutic value in cancer.

Significant efforts have been made thus far to develop HER2-targeted therapies for cancer. The humanized monoclonal antibody (mAb) trastuzumab (Herceptin^®^ Genentech South San Francisco, CA, USA) was the first FDA-approved HER2-targeted therapy for HER2-positive metastatic breast cancer in 1998 and has been expanded to treat HER2-positive gastric and gastroesophageal cancer [30]. Trastuzumab was derived from the murine anti-HER2 mAb 4D5, which binds to the extracellular domain of HER2, thereby blocking HER2 signaling and inducing antibody-dependent cellular cytotoxicity (ADCC) and antibody-dependent cellular phagocytosis (ADCP) [31,32]. Currently, trastuzumab is the standard of care treatment administered in combination with chemotherapy, radiotherapy, or other targeted therapies. Unfortunately, a substantial number of patients gradually develop resistance to the treatment, resulting in limited benefits and disease relapse [33,34]. Therefore, a second humanized anti-HER2 mAb, pertuzumab, which recognizes a different epitope in the extracellular domain of HER2, was approved [2]. The combination of the two mAbs with chemotherapy exhibited superior results to monotherapy and to transtuzumab plus chemotherapy combination therapy, indicating an additive efficacy [35,36]. Small molecules that target the HER2 intracellular tyrosine kinase domain for blockage of phosphorylation events and downstream signaling have also been approved for clinical use. Lapatinib, a small molecule that reversibly inhibits both HER1 and HER2 tyrosine kinases, was found to be effective for HER2-positive metastatic breast cancer patients that had failed trastuzumab-based first-line treatments [37,38,39]. The irreversible small-molecule tyrosine kinase inhibitor (TKI) neratinib (HKI-272), which targets HER1, HER2, and HER4, improved the survival rates and treatment outcomes in HER2-positive breast cancer patients. Currently, neratinib is approved as a third-line treatment option [40,41]. Additionally, antibody–drug conjugates (ADCs) are being developed to target HER2-positive tumor cells. Ado-trastuzumab emtansine (T-DM1), trastuzumab linked to the chemotherapy anti-microtubule agent emtansine, was the first approved ADC therapy as a second-line treatment option for HER2-positive breast cancer. T-DM1 treatment exhibited a longer invasive disease-free survival (DFS), a higher overall response rate (ORR), and a lower risk of recurrence than trastuzumab and showed efficacy in a subset of breast cancer patients with brain metastases [42,43]. A new generation of ADCs, fam-trastuzumab deruxtecan-nxki (T-DXd), which combines trastuzumab with the topoisomerase inhibitor MAAA-1181a, dramatically improved ORR, DFS, and overall survival in HER2-positive metastatic breast cancer patients, demonstrated superior efficacy than T-DM1, and received accelerated FDA approval for treating HER2-positive breast cancer patients who have received two or more prior HER2-targeted regimens [44]. Despite these encouraging results in the clinic, the development of resistance to HER2-targeted therapies is disturbing [45], and drug-related adverse events are common and grave [46]. More potent, durable, and safe HER2-targeted therapies are desperately needed for HER-positive cancer patients.

In this study, we developed a CD3 bispecific antibody (bsAb) that specifically targets HER2-positive tumor cells. The bsAb functions as a bridge connecting T cells to tumor cells, mimicking the T-cell-receptor–tumor antigen peptide/histocompatibility complex interaction to stimulate T cell activation and antitumor function [47]. Bispecific antibodies have emerged as a promising treatment for hematological malignancies such as acute myeloid leukemia and multiple myeloma, as well as solid tumors, including lung cancer and colorectal cancer [48,49]. We and others have previously shown that the delivery of synthetic messenger RNA (mRNA) instead of the antibody triggers potent and long-lasting antitumor immunity [50,51,52,53]. Here, we generated a HER2-CD3 bsAb with a mutated Fc (P329G, L234A, and L235A) domain to reduce Fc effector functions [53,54] and leveraged lipid nanoparticle (LNP) technology to achieve efficient delivery of the bsAb-encoding mRNA. Our data show that LNP-mediated delivery of the HER2-CD3-Fc mRNA resulted in the robust and sustained expression of the HER2-CD3-Fc bsAb in various types of cells, which induced a potent and highly specific T-cell-dependent antitumor effect against multiple HER2-positive tumor cells in vitro. Intratumoral injection of the bsAb mRNA-LNPs completely blocked the HER2-positive A1847 tumor growth in a mouse xenograft model of human ovarian cancer. Thus, the HER2-CD3-Fc mRNA-LNPs holds promise as a novel and effective therapy for HER2-positive cancer.

## 2. Materials and Methods

### 2.1. Cell Lines and Antibodies

A549, A1847, SKOV-3, and SKBR3 cells were cultured in Dulbecco’s modified Eagle’s medium (DMEM, Gibco, Thermo Fisher, Waltham, MA, USA) with 10% fetal bovine serum (FBS, Gibco) and penicillin/streptomycin. MDA-MB-231, NCI-H460, and 4T1 cells were cultured in RPMI1640 medium with 10% FBS and penicillin/streptomycin. HEK293S cells were cultured in FreeStyle™ F17 medium with GlutaMAX™ (Gibco) and Pluronic™ F-68 nonionic surfactant (Gibco). A549 and A1847 *her2*^−/−^ knockout cell lines were generated by electroporation with *her2* sgRNAs (CAUAGUUGUCCUCAAAGAGC, AACAAUACCACCCCUGUCAC, and CGCUCACAACCAAGUGAGGC) and Cas9 protein (all from Synthego, Redwood City, CA, USA) using the Neon system (Thermo Fisher). 4T1-HER2-ECD and 4T1-HER2-ECD-IV cells were generated by transduction with lentiviruses collected from HEK293FT cells co-transfected with the pPACKH1 lentivector packaging mix (System Biosciences, Palo Alto, CA, USA) and a 3rd-generation lentiviral vector encoding the extracellular domain and transmembrane region of human HER2 (HER2-ECD, aa 1–675) or a truncated HER2-ECD with the epitope-containing subdomain IV, the transmembrane region, and a short intracellular tail (HER2-ECD-IV, aa 509–700) [55]. Human peripheral blood mononuclear cells (PBMCs) were isolated from LRS chambers obtained from the Stanford Hospital Blood Center (Institutional Review Board-approved protocol #13942) by sedimentation over Ficoll-Paque (GE Healthcare, Chicago, IL, USA) [56]. PBMCs were suspended at 1 × 10^6^ cells/mL in AIM-V medium (Thermo Fisher) containing 10% FBS and 10 ng/mL recombinant human IL-2 (Thermo Fisher), activated with CD3/CD28 Dynabeads (Thermo Fisher) at a 1:1 ratio, and cultured for 10–12 days, with fresh medium being added every 2–3 days. All cell lines and primary cells were cultured in a humidified 5% CO_2_ incubator. APC anti-human IgG Fc antibody (#410712), APC anti-human CD3 antibody (#317318), and 7-AAD viability staining solution (#420404) were from Biolegend (San Diego, CA, USA). HRP anti-human IgG Fc antibody (#A0170) was from Sigma (Burlington, MA, USA).

### 2.2. Design and Cloning of the Bispecific Antibody Construct

The HER2-CD3-Fc bsAb contains a single-chain variable fragment (scFv) derived from the humanized anti-HER2 4D5 mouse mAb [57], a scFv derived from the anti-human CD3 mAb [58], and the human IgG1 Fc domain containing L234AL235A (LALA) and P329G mutations to block undesired Fc-dependent immune responses (Figure 1A) [54,59]. To generate the HER2-mCD3-Fc bsAb recognizing mouse CD3, the human CD3 scFv sequence was replaced with a mouse CD3 scFv [60]. Each scFv contains the variable fragments of the parent Ab’s heavy chains and light chains with a 3×(GGGGS) linker added between the two fragments. The complete amino acid sequence is shown in Appendix A. The bsAb DNA sequences were cloned into a DNA vector with a T7 promoter, a 5′ untranslated region (UTR), a 3′UTR, and a 150-base pair poly A tail, as described [51,53]. The constructs were verified by Sanger sequencing.

### 2.3. In Vitro Transcription

The bsAb mRNA was transcribed in vitro from the DNA templates using the HiScribe^®^ T7 mRNA Kit (NEB, Ipswich, MA, USA) with CleanCap^®^ Reagent AG (NEB, #E2080) following the manufacturer’s protocol. Briefly, 1 μg of linearized DNA template; 0.5 × Reaction Buffer; 4 mM of CleanCap Reagent AG; 5 mM of ATP, CTP, GTP, and methylpseudo-UTP (Trilink, San Diego, CA, USA, #N-1081-100); and T7 RNA polymerase were mixed and incubated for 2 h at 37 °C. DNAse I treatment was performed for 15 min at 37 °C to degrade the template DNA. The mRNA was purified with Monarch RNA Cleanup Kit (NEB, #T2050), according to the manufacturer’s protocol. Each mRNA was validated for proper size and stability using agarose gel electrophoresis.

### 2.4. Preparation of bsAb mRNA-Lipid Nanoparticles and Transfection of Target Cells

To prepare the mRNA-LNPs, an aqueous solution of mRNA in 100 mM sodium acetate (pH 4.0) (Thermo Fisher) was mixed with a lipid complex containing SM-102 (Cayman Chemical, Ann Arbor, MI, USA, #33474), DSPC (Avanti Polar Lipids, Alabaster, AL, USA, #850365P), cholesterol (Sigma, #C8667), and DMG-PEG2000 (Cayman Chemical, #33945) (molar ratio of 100:20:77:3 in ethanol) at a flow rate ratio of 3:1 (mRNA:lipids) using the PreciGenome Flex S System (San Jose, CA, USA). The mRNA-LNPs were dialyzed against sterile PBS and concentrated using Amicon^®^ Ultra-15 centrifugal filter units (Millipore, Sigma, #UFC9100). The particle size and polydispersity index (PDI) of the mRNA-LNPs were detected using an Anton Paar Litesizer 500 System (Anton Paar, Ashland, VA, USA). The HER2-CD3-Fc mRNA-LNPs have a mean hydrodynamic diameter of 92.7 nm ± 2.8 nm and a mean PDI of 0.075 ± 0.063.

For transfection, mRNA-LNPs containing 1 μg of bsAb mRNA were added to 0.5–1.0 × 10^6^ cells in a dropwise manner. For analysis of the bsAb expression kinetics, a portion of culture medium from HEK293S cells was collected at indicated time points post-transfection. For other analyses, culture medium containing secreted bsAb was collected 72 h post-transfection. The media were centrifuged at 300× *g* for 5 min and the supernatants were used.

### 2.5. Sodium Dodecyl Sulfate Polyacrylamide Gel Electrophoresis (SDS-PAGE) and Western Blotting

Medium containing secreted bsAb was mixed with SDS loading buffer and incubated at 95 °C for 5 min. The samples were loaded on a NuPAGE™ 4~12%, Bis-Tris, 1 mm, Mini Protein Gel (Invitrogen, Thermo Fisher, #NP0322BOX) and run in 1× NuPAGE™ MOPS SDS Running Buffer (Invitrogen, #NP000102) at 100 V for 75 min. For SDS-PAGE, the gel was stained with SimplyBlue™ SafeStain (Invitrogen, #LC6065) at room temperature (RT) overnight and destained with water before imaging. For Western blotting, proteins were transferred to Sequi-Blot™ polyvinylidene difluoride membranes (Bio-Rad, Hercules, CA, USA, #1620184). The membranes were blocked in Tris-buffered saline containing 5% nonfat milk powder and Tween-20 (TBST; 140 mM NaCl, 10 mM Tris/HCl, pH 7.4, and 0.1% Tween-20) for 1 h at RT and incubated with HRP-conjugated anti-human IgG Fc antibody overnight at 4 °C. After rigorous washing, the membranes were incubated with the Pierce™ ECL Western Blotting Substrate (Invitrogen, #32209) and visualized using the ImageQuant™ LAS 500 system (GE Healthcare). To quantify HER2-CD3-Fc bsAb secretion by Western blotting, serially diluted human IgG standards with known protein concentrations were used. Image J software (version 1.54i, National Institutes of Health, Bethesda, MD, USA) was used to quantify the band intensities.

### 2.6. Flow Cytometry

To measure the binding of the bsAbs to target cells, 0.25 × 10^6^ cells were washed with PBS containing 2 mM EDTA and 0.5% BSA (washing buffer) and incubated with 0.1 mL of 1:10 diluted bsAb-containing medium in washing buffer for 30 min on ice. The cells were then washed and suspended in 0.1 mL of washing buffer supplemented with 1 µL of APC-conjugated anti-human Fc antibody and 1 µL of 7-AAD solution. After 30 min of incubation on ice, the cells were washed and analyzed on a FACSCalibur flow cytometer (BD Biosciences, Fremont, CA, USA). The 7-AAD-positive dead cells were excluded by gating. The data were processed using the software FlowJo 7.6 (BD Biosciences).

### 2.7. Real-Time Cell Analysis (RTCA)

Target cells were seeded in triplicate into 96-well E-plates (Agilent Technologies, Santa Clara, CA, USA, #300600900) at a density of 1–4 × 10^4^ cells per well and monitored overnight for monolayer impedance using the xCELLigence real-time cell analysis system (Agilent Technologies). The next day, the medium was removed and replaced with medium containing effector T cells at a 10:1 effector:target ratio, either alone or with serially diluted bsAb-containing medium or with bsAb mRNA-LNPs. The target cells were monitored for at least an additional 24 h, and monolayer impedance was plotted over time. At the end of the experiment, the medium was collected and centrifuged to remove cells. Cytotoxicity was calculated using the following formula: ((*cell index of tumor cells*) − (*cell index of tumor cells plus different treatment*))/(*cell index of tumor cells*) *× 100%.*

### 2.8. Enzyme-Linked Immunoassay (ELISA)

The medium collected from the real-time cytotoxicity assay was analyzed for human granzyme B, IFN-λ, and TNF-α secretion levels using Quantikine kits (R&D Systems, Minneapolis, MN, USA), according to the manufacturer’s protocols. Briefly, 96-well microplates were coated with capture antibodies overnight at 4 °C, then incubated with the collected medium at RT for 2 h. After three washes, the microplates were incubated with detection antibodies for 2 h, and subsequently with streptavidin-HRP for 30 min. Following incubation with substrate solution for 20 min, the peroxidase reactions were stopped with sulfuric acid and the plates were read at 450 nm. The cytokine concentration was calculated by interpolation using the kits’ standards.

### 2.9. Mouse Xenograft Tumor Model

Six-week-old NOD/SCID/IL2rγ^null^ (NSG) mice (Jackson Laboratories, Bar Harbor, ME, USA) were housed and handled in strict accordance with the Institutional Animal Care and Use Committee (IACUC) guidelines and protocol (#LUM-001). A total of 2 × 10^6^ A1847 tumor cells in a volume of 100 µL were implanted subcutaneously on day 0. On day 7, the mice were randomly assigned into three groups and injected intratumorally with 20 µL sterile PBS, 20 µL HER2-CD3-Fc mRNA-LNPs (1 µg mRNA), or 20 µL GFP mRNA-LNPs (1 µg mRNA). Intratumoral injections were repeated on days 14 and 21. On day 9, 1 × 10^7^ human T cells were injected intravenously. The tumor size was measured with calipers every 3 days, and the tumor volume was calculated using the formula (*Length × Width*^2^)/2. The mouse body weight was also measured every 3 days. Blood samples were collected at the end of the study for analysis of the toxicology biomarkers alanine aminotransferase (ALT), aspartate aminotransferase (AST), and amylase at IDEXX BioAnalytics (Columbia, MO, USA).

### 2.10. Statistical Analysis

To compare the differences between two groups or treatments, an unpaired Student’s *t* test was applied. To compare the differences between three or more groups or treatments, a two-way ANOVA with multiple comparisons was performed. GraphPad Prism 7.0 software was used for all data analysis. Data are presented as mean ± standard deviation (SD). A *p* value < 0.05 was considered statistically significant (* *p* < 0.05, ** *p* < 0.01, *** *p* < 0.001, and **** *p* < 0.0001). A *p* value ≥ 0.05 was statistically not significant (N.S.).

## 3. Results

### 3.1. Design, Expression, and Binding Activity of the HER2-CD3-Fc Bispecific Antibody

To generate the HER2-CD3-Fc bsAb, we used the humanized mouse anti-HER2 4D5 mAb [57] for the design of the HER2 single chain variable fragment (scFv) and the anti-human CD3 mAb [58] for the CD3 scFv. The variable fragments of heavy chains (VHs) and light chains (VLs) were tandemly connected with a 3×(GGGGS) linker. The human IgG1 Fc domain was attached to the C terminus of the bsAb to enhance its stability and form homodimers. L234A, L235A, and P329G mutations were introduced in the Fc domain to reduce its binding to IgG Fc receptors FcγRI/FcγRII/FcγRIII and complement component C1q, thereby reducing the bsAb’s ability to trigger ADCC and ADCP and complement activation in vivo (Figure 1A) [54,59]. We produced the HER2-CD3-Fc bsAb mRNA by in vitro transcription and prepared LNPs containing the mRNA.

To assess the expression of the bsAb, we transfected HEK293S suspension cells with the HER2-CD3-Fc mRNA-LNPs and collected the culture medium containing the secreted bsAb. SDS-PAGE revealed a dominant band at approximately 160 kDa under non-reducing conditions and a dominant band of approximately 80 kDa under reducing conditions (Figure 1B), confirming the expression of the bsAb as a full-sized dimer. Western blotting using an anti-human IgG Fc antibody confirmed the bsAb size and Fc domain (Figure 1C). To explore the kinetics of expression, we measured the bsAb concentrations at various time points after transfection using Western blotting. The bsAb level steadily increased after transfection, peaked at 72–96 h, and remained high at 168 h (Figure 1D,E). The transfection of 1 × 10^6^ HEK293S cells with 1 µg of mRNA produced about 9 µg of secreted bsAb at 72 h and 7.5 µg at 168 h, indicative of the bsAb’s high expression and stability.

To analyze bsAb binding to HER2 and CD3, we performed flow cytometry with various HER2-positive cell lines and human T cells using the medium from transfected HEK293S cells and an APC-conjugated anti-human IgG Fc secondary antibody. The bsAb stained HER2-positive cancer cell lines, including human non-small-cell lung cancer (NSCLC) cell lines A549 and NCI-H460, human ovarian cancer cell lines SKOV-3 and A1847, and human breast cancer cell line SKBR3, but not HER2-negative Chinese hamster ovary (CHO) cells (Figure 1F) [61]. Moreover, the bsAb stained CD3-positive human T cells (Figure 1F). We also tested the binding of the bsAb to MDA-MB-231, a triple-negative breast cancer (TNBC) cell line (negative for the estrogen receptor, progesterone receptor, and HER2) with a low HER2 level [62,63,64,65]. Importantly, we found that the HER2-CD3-Fc bsAb also binds to MDA-MB-231cells (Figure 1F), indicative of the bsAb’s high binding affinity for HER2.

To test whether the HER2-CD3-Fc mRNA-LNPs can transfect and induce bsAb secretion from cancer cells, we added the mRNA-LNPs to the ovarian cancer cells SKOV-3 and A1847. Western blotting analysis of the culture medium demonstrated expression and secretion of the bsAb by both cancer cell lines, albeit at significantly lower levels compared to HEK293S cells (Figure 1G). Transfection of 1 × 10^6^ A1847 cells with 1 µg of mRNA produced about 0.6 µg of secreted bsAb at 72 h, whereas 1 µg of mRNA generated only 5 ng of secreted bsAb from the SKOV-3 cells. This result indicates that the HER2-CD3-Fc mRNA-LNPs induce varying levels of bsAb expression and secretion across different target cells. The secreted bsAb from the A1847 cells was able to bind to both T cells and SKOV-3 cells (Appendix A), indicating that the bsAb produced by transfected HER2-positive tumor cells retains antigen-binding activity. Similarly, we transfected the mRNA-LNPs into the breast cancer cells SKBR3 and verified that the produced bsAb was also able to bind to both T cells and SKOV-3 cells (Appendix A). Collectively, these results show that the mRNA-LNPs can transfect various types of target cells to produce the HER2-CD3-Fc bsAb that binds to both HER2-positive tumor cells and human T cells.

**Figure 1 vaccines-12-00808-f001:**
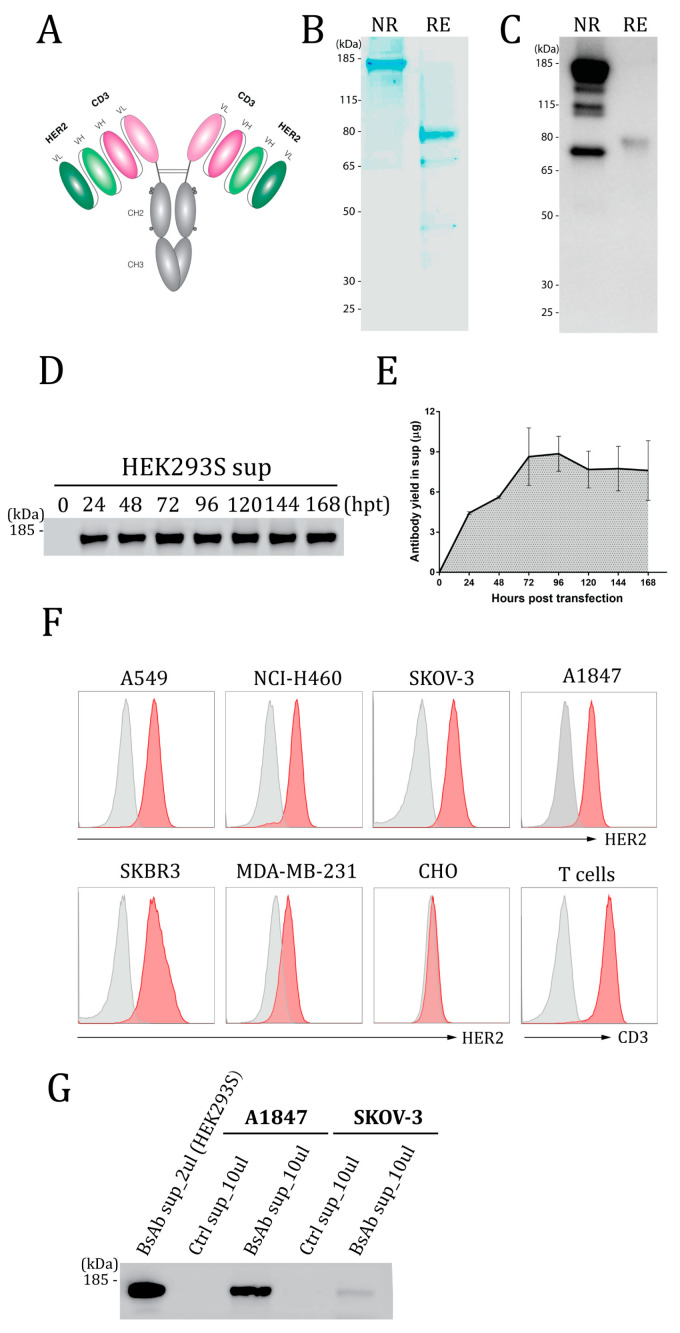
Design and expression of the HER2-CD3-Fc bispecific antibody (bsAb). (**A**) Structure of the HER2-CD3-Fc bsAb. The bsAb has a symmetrical structure consisting of two identical monomers linked through disulfide bonds (two horizon lines). Each monomer contains a HER2 scFv, a CD3 scFv, and a human Fc domain with three amino acid mutations (L234A, L235A, and P329G, denoted by three dots). VL, light chain variable domain; VH, heavy chain variable domain; CH2, heavy chain constant domain 2; CH3, heavy chain constant domain 3. Each scFv consists of a VH and a VL, whereas the Fc domain consists of CH2 and CH3. (**B**–**E**,**G**) Expression of the HER2-CD3-Fc bsAb. A total of 1 × 10^6^ HEK293S (**B**–**E**,**G**), A1847 (**G**) or SKOV-3 (**G**) cells were transfected with 20 µL of LNPs containing 1 µg of bsAb mRNA, and the culture medium was collected at 48 h post transfection (hpt) (**B**,**C**) or every 24 h until 168 hpt (**D**,**E**) or at 72 hpt (**G**). The medium was subjected to SDS-PAGE (B) or Western blotting (**C**,**D**,**G**) under reducing (RE, with 2-Mercaptoethanol) conditions or non-reducing (NR, without 2-Mercaptoethanol) conditions. For SDS-PAGE, the gel was stained with SimplyBlue SafeStain buffer. For Western blotting, the samples were stained with HRP-conjugated anti-human Fc antibody. The gels in (**D**,**G**) were run under non-reducing conditions. Quantification of the bsAb was based on serially diluted human IgG standards. (**E**) is the quantification plot of the bsAb bands in (**D**). Data in (**D**) are representative of three independent experiments, and the data in (**E**) are presented as the means ± SD. (**F**) Binding analysis of the HER2-CD3-Fc bsAb by flow cytometry. A total of 1 × 10^6^ HEK293S cells were transfected with 20 µL of LNPs containing 1 µg of the bsAb mRNA for 72 h. The medium was then collected and used at a dilution of 1:10 to stain human T cells and cell lines, including A549, NCI-H460, SKOV-3, A1847, SKBR3, MDA-MB-231, and CHO (red histograms). Non-transfected HEK293S cell medium was used at 1:10 as a negative control (gray histograms). After rinsing, the samples were stained with APC anti-human IgG Fc. Dead cells were excluded from analysis with the 7-AAD viability dye.

### 3.2. The HER2-CD3-Fc bsAb Induces Potent T Cell Cytotoxicity against HER2-Positive Tumor Cells

To determine whether the bsAb could induce human T cells to kill HER2-positive tumor cells, we co-incubated effector T cells with different target cells in the presence or absence of serially diluted HEK293S medium containing the secreted HER2-CD3-Fc bsAb and monitored target cell viability using the impedance-based xCELLigence real-time cell analysis (RTCA) system. The HER2-CD3-Fc bsAb significantly enhanced T cell cytotoxicity against the NSCLC cell lines A549 and NCI-H460 in a dose-dependent manner; furthermore, in the absence of T cells, the bsAb did not affect the tumor cells (Figure 2A,B). Notably, a high concentration of the bsAb induced the T cells to kill almost 100% of the tumor cells. In addition, the HER2-CD3-Fc bsAb strongly mediated T-cell-dependent killing of other HER2-positive tumor cells, including the ovarian cancer cell lines SKOV-3 and A1847 and the breast cancer cell lines SKBR3 and MDA-MB-231 (Figure 2C–F). In contrast, the bsAb failed to induce T-cell killing of HER2-negative CHO cells (Figure 2G). These results indicate that the HER2-CD3-Fc bsAb induces potent and specific T cell cytotoxicity against HER2-positive tumor cells.

### 3.3. The HER2-CD3-Fc bsAb Induces T Cells to Secrete Cytokines against HER2-Positive Tumor Cells

To analyze whether the bsAb mediated T cell activation in response to the tumor cells, we measured the levels of the effector molecules granzyme B, IFN-λ, and TNF-α secreted into the medium by T cells. The HER2-CD3-Fc bsAb in the culture significantly enhanced T cell secretion of IFN-λ, TNF-α, and granzyme B in response to NCI-H460 tumor cells in a dose-dependent manner (Figure 3A–C). In contrast, in the absence of the bsAb, the T cells produced low and variable levels of IFN-λ, granzyme B, and TNF-α. Similarly, the bsAb strongly promoted T cell secretion of IFN-λ, TNF-α, and granzyme B in response to SKOV-3 cells (Figure 3D–F). These results suggest that the T cells were functionally activated by the bsAb to kill the tumor cells. We also tested T cell secretion of IFN-λ and granzyme B during co-culture with other HER2-positive tumor cells, including A549, A1847, SKBR3, and MDA-MB-231 cells, and consistently observed a dose-dependent induction of IFN-λ and granzyme B by the bsAb (Figure 3G–N). In contrast, the bsAb failed to induce T cell secretion of IFN-λ and granzyme B against HER2-negative CHO cells (Figure 3O,P). Collectively, the data indicate that the HER2-CD3-Fc bsAb strongly induces T-cell-dependent secretion of cytokines and effector molecules against HER2-positive tumor cells.

### 3.4. The HER2-CD3-Fc mRNA-LNPs Induce Potent T Cell Cytotoxicity against HER2-Positive Tumor Cells

Since the HER2-CD3-Fc bsAb induces strong T-cell-dependent killing of HER2-positive tumor cells, we tested whether T cells would kill tumor cells transfected directly with the HER2-CD3-Fc mRNA-LNPs. We co-incubated T cells with the NSCLC cells A549 in the presence or absence of serially diluted HEK293S medium containing the HER2-CD3-Fc bsAb or different amounts of the HER2-CD3-Fc mRNA-LNPs and monitored A549 cell growth using RTCA. The mRNA-LNPs induced the robust killing of the tumor cells in a dose-dependent manner, similar to the HER2-CD3-Fc bsAb, but with a delayed onset of action (Figure 4A). This effect also occurred with a different target cell line, the ovarian cancer cells A1847 (Figure 4B). The delayed response from the mRNA-LNPs likely stems from the extra time required for the mRNA to be translated and the bsAb to be secreted from the transfected cells. Regardless, T cells were potently activated by the mRNA-LNPs and secreted high levels of IFN-λ and granzyme B, which were even higher than the levels induced by the HER2-CD3-Fc bsAb (Figure 4C–F). Similarly, the mRNA-LNPs strongly promoted T-cell-dependent killing (Appendix A) and secretion of IFN-λ, TNF-α, and granzyme B (Appendix A) against other HER2-positive tumor cells, including NCI-H460, SKOV-3, and MDA-MB-231. These results indicate that the HER2-CD3-Fc mRNA-LNPs can be directly used in place of the HER2-CD3-Fc bsAb to induce T cell cytotoxicity against HER2-positive tumor cells.

### 3.5. The HER2-CD3-Fc bsAb Mediates a Highly Specific Antitumor Effect Dependent on Both HER2 and CD3

To assess the strict dependence of the HER2-CD3-Fc bsAb-induced T cell cytotoxicity on the HER2 antigen and exclude the possibility of a non-specific antitumor effect of the bsAb, we generated isogeneic *her2* knockout cells using the CRISPR/Cas9 system for both the NSCLC cell line A549 and the ovarian cancer cell line A1847. Both cell lines originally expressed a high level of HER2 on the cell surface (Figure 1F) and were readily subjected to the HER2-CD3-Fc bsAb-mediated T-cell killing (Figure 2). The HER2-CD3-Fc bsAb recognized the parental A549 and A1847 cells well but failed to bind to the *her2* knockout cells (Figure 5A and Appendix A). As a result, the A549 *her2* knockout cells became resistant to HER2-CD3-Fc bsAb-mediated T cell cytotoxicity (Figure 5C), unlike parental A549 cells (Figure 5B). Similarly, the HER2-CD3-Fc bsAb induced potent T-cell killing of parental A1847 tumor cells, but not A1847 *her2* knockout tumor cells (Appendix A). These results indicate that the bsAb triggers T-cell killing of tumor cells in a manner dependent on its binding to HER2 on the tumor cells. Consistently, the loss of HER2 on the knockout tumor cells led to dramatically reduced T cell secretion of IFN-λ and granzyme B in the presence of the HER2-CD3-Fc bsAb (Figure 5D,E and Appendix A), indicating that HER2 binding was required for the bsAb-mediated cytotoxic T cell activation.

To determine whether the *her2* knockout cells were also resistant to HER2-CD3-Fc mRNA-LNP-mediated T cell cytotoxicity, we repeated the RTCA with the HER2-CD3-Fc mRNA-LNPs in lieu of the bsAb. The HER2-CD3-Fc mRNA-LNPs induced potent T cell cytotoxicity against the parental A549 and A1847 cells but failed to facilitate the killing of the *her2* knockout cells (Appendix A). Consequently, the T cells released very low levels of IFN-λ and granzyme B against the *her2* knockout cells (Appendix A).

To further assess the HER2-CD3-Fc bsAb-mediated HER2-specific antitumor effect, we generated the 4T1.2 (4T1) mouse mammary cancer cell line stably expressing the extracellular domain and transmembrane region of human HER2 (HER2-ECD, aa 1–675) (4T1-HER2-ECD) (Figure 5F) [55]. The HER2-CD3-Fc bsAb bound readily to these 4T1-HER2-ECD cells, but not to the parental 4T1 cells (Figure 5G), indicating that the antigen HER2-ECD was well expressed and folded on the mouse tumor cells. Interestingly, following co-culture with human T cells, the HER2-CD3-Fc bsAb induced potent killing of 4T1-HER2-ECD cells, but elicited no obvious cytotoxic effect on parental 4T1 cells (Figure 5H,I). Consequently, the T cells secreted high levels of IFN-λ and granzyme B in response to 4T1-HER2-ECD cells, but not to the parental 4T1 cells (Figure 5K,L,N,O). These results further confirm the HER2-dependent antitumor effect of the bsAb.

The HER2 extracellular domain contains four subdomains, including two tandem receptor-L regions (subdomains I and III), a furin-like cysteine rich region (subdomain II), and a growth factor receptor region (subdomain IV) containing the HER2-CD3-Fc bsAb epitope (Figure 5F) [66]. To determine whether the epitope-containing subdomain IV alone is sufficient to trigger the antitumor activity of the HER2-CD3-Fc bsAb, we generated 4T1 cells stably expressing a truncated version of human HER2-ECD containing only subdomain IV and the transmembrane region with an extra-short intracellular tail (HER2-ECD-IV, aa 509–700) (4T1-HER2-ECD-IV) [55]. Again, the HER2-CD3-Fc bsAb efficiently bound to the 4T1-HER2-ECD-IV cells (Figure 5G) and induced robust T cell cytotoxicity (Figure 5J) and IFN-λ/granzyme B secretion (Figure 5M,P) against the subdomain-IV-expressing tumor cells. Therefore, the epitope-containing subdomain IV of HER2 is sufficient for the induction of the HER2-CD3-Fc bsAb-mediated antitumor effect.

Additionally, to confirm the CD3-dependent antitumor activity of the HER2-CD3-Fc bsAb, we replaced the human CD3 scFv with a mouse CD3 svFv and generated the HER2-mCD3-Fc bsAb. The HER2-mCD3-Fc bsAb also bound strongly to the 4T1-HER2-ECD cells and the 4T1-HER2-ECD-IV but failed to recognize human T cells (Figure 5G). As expected, the ablation of CD3 binding abolished the HER2-mCD3-Fc bsAb’s ability to trigger human T-cell-dependent cytotoxicity (Figure 5I,J) and IFN-λ/granzyme B secretion (Figure 5K–P) against the HER2-overexpressing tumor cells, suggesting that the HER2-CD3-Fc bsAb relies on CD3 binding to recruit T cells and elicit antitumor activity.

Together, these results indicate that the HER2-CD3-Fc bsAb mediates a highly specific antitumor effect that is strictly dependent on its binding to HER2 and CD3.

### 3.6. The HER2-CD3-Fc mRNA-LNPs Induce a Potent Antitumor Effect In Vivo

Since the HER2-CD3-Fc mRNA-LNPs exhibited potent T-cell-dependent killing of HER2-positive tumor cells in vitro, we examined whether the mRNA-LNPs would exert an antitumor effect against A1847 tumor cells in the NSG mouse xenograft model [67]. A total of 2 × 10^6^ A1847 cells were inoculated into NSG mice subcutaneously (s.c.) on day 0 of the study. PBS, GFP mRNA-LNPs, or HER2-CD3-Fc mRNA-LNPs were subsequently injected intratumorally (i.t.) on days 7, 14, and 21, respectively. On day 9, 1 × 10^7^ human T cells were infused into the mice intravenously (i.v.). The tumor volume and body weight of the mice were measured every three days (Figure 6A). The tumors grew exponentially in the PBS and the GFP mRNA-LNP treatment groups; however, in the HER2-CD3-Fc mRNA-LNP treatment group, the tumors started to shrink on day 14 and completely disappeared by day 28 (Figure 6B,C). Therefore, the HER2-CD3-Fc mRNA-LNPs, but not the GFP mRNA-LNPs, significantly blocked the growth of the A1847 tumors in vivo. The body weights of the mice were comparable between the groups (Figure 6D), and the toxicology biomarkers alanine aminotransferase (ALT), aspartate aminotransferase (AST), and amylase in the blood samples of the treated mice showed no obvious increase (Figure 6E–G) [68], suggesting no toxicity associated with the mRNA-LNP treatments. In addition, no obvious physical or behavioral changes were observed in these mice throughout the study. We conclude that intratumoral injection of the HER2-CD3-Fc mRNA-LNPs induces a strong antitumor response against HER2-positive xenograft tumors.

## 4. Discussion

HER2 is a tumor antigen widely expressed and dysregulated in various cancers of epithelial origin, including breast cancer, ovarian cancer, and NSCLC [16,19]. Targeting HER2 for the treatment of HER2-positive cancers has been efficacious and promising for advanced or metastatic breast cancer, gastric cancer, and gastroesophageal cancer [30,37,38,39,42]. Despite the unprecedented success, emerging drug resistance and severe side effects limit the long-term efficacy of HER2-targeted therapies for the HER2-positive cancer patients [34,43,69]. Additional efficacious and durable therapies are desperately needed.

In this study, we developed a HER2-targeted CD3 bsAb-based therapy designed for intratumoral delivery using lipid nanoparticles (LNPs). We, for the first time, demonstrate that transfection of tumor cells with the HER2-CD3-Fc mRNA-LNPs results in robust and sustained bsAb secretion. The secreted HER2-CD3-Fc bsAb binds to both HER2 and CD3 and induces potent T cell cytotoxicity and cytokine secretion against multiple HER2-positive tumor cells, including MDA-MB-231 cells with low HER2 expression. The bsAb-mediated antitumor effect is highly specific and strictly dependent on its binding to the HER2 epitope and CD3, and tumor cells lacking HER2 become resistant to the bsAb-induced T cell cytotoxicity. The bsAb also requires binding to CD3 to recruit T cells, and ablation of CD3 binding abrogates the bsAb’s ability to elicit antitumor activity. Intratumoral injection of HER2-CD3-Fc mRNA-LNPs triggers a strong antitumor response against HER2-positive tumors and completely blocks the A1847 ovarian cancer xenograft tumors in vivo without showing any signs of toxic effects. These results establish the HER2-CD3-Fc mRNA-LNPs as a potent, safe, and promising HER2-targeted therapy that is worth clinical evaluation.

Therapeutic antibodies have emerged as a promising and predominant class of new drugs for cancer treatment, owing to their specific mechanism of actions and reduced side effects [70]. In contrast to mAbs, bsAbs engage two different targets and offer more specific and synergistic therapeutic effects for complex diseases like cancer [71,72]. We generated a HER2-CD3-Fc bsAb in this study for the treatment of HER2-positive tumors. Compared with trastuzumab, our HER2-CD3-Fc bsAb has a CD3 scFv that recruits and activates effector T cells to attack HER2-positive tumor cells and, thus, potentially offers better and more potent antitumor effects [30,32,73]. Our in vitro and in vivo data consistently show that the HER2-CD3-Fc bsAb induces potent cytotoxicity against various types of HER2-positive tumor cells in a manner that is both T-cell-dependent and HER2-specific. In all of the different types of HER2-positive tumors we tested, which include NSCLC, ovarian, and breast cancers, we consistently observed the near-complete killing of tumor cells. This indicates that the bsAb has the potential to treat all types of HER2-positive cancers. Importantly, we found that the bsAb binds to and triggers a strong antitumor effect against HER2-low TNBC MDA-MB-231 cells. TNBC, characterized by the lack of an estrogen receptor, progesterone receptor, and HER2, is an extremely heterogenous and aggressive cancer that does not respond to available HER2-targeted therapies, lacks effective treatment regimens, and is associated with a high chance of metastasis and poor prognosis [74]. A recent phase III clinical trial evaluating the efficacy of the trastuzumab-based ADC drug T-DXd for treating HER2-low advanced breast cancer, including TNBC, found that the drug is more effective than chemotherapy and significantly prolongs patients’ progression-free and overall survival [75]. Therefore, it is very possible that our HER2-CD3-Fc bsAb is also effective in treating HER2-low advanced breast cancer.

mRNA has gained growing interest and shows tremendous potential in a variety of therapeutic applications, including viral vaccines, cancer immunotherapies, and gene therapies [76,77,78]. Two key challenges related to mRNA-based therapies are to maintain mRNA stability and to achieve targeted delivery. LNPs have been shown to be a safe, reliable, and efficient mRNA delivery platform that addresses these challenges effectively since its application in the two widely used COVID-19 mRNA vaccines mRNA-1273 [79] and BNT162b [80]. In this study, we used the LNP platform for the efficient delivery of our HER2-CD3-Fc mRNA. We demonstrate that the LNPs can deliver the bsAb mRNA efficiently to various types of cell lines, resulting in high and sustained expression of active antibodies. Moreover, LNP-mediated delivery of the bsAb mRNA exhibits a robust antitumor effect against multiple HER2-positive tumor cells and induces higher levels of IFN-λ and granzyme B than the direct application of the bsAb. The HER2-CD3-Fc mRNA-LNPs induce a peak bsAb secretion at 72–96 h and maintain a high antibody level, even at 168 h, in the culture of transfected HEK293S cells. This indicates that the LNP delivery of bsAb mRNA is more advantageous than the direct application of bsAbs in inducing a strong and durable antitumor effect, which has significant implications in clinical settings. Antibody drugs are predominantly administered via intravenous infusion in the clinic, which results in low penetration in solid tumors, rapid clearance, and a very short antibody half-life of 1–4 days [81,82,83]. To maintain the desired therapeutic effects, multiple infusions with shorter dosing intervals are usually required. In addition, intravenous infusion directly exposes key organs and tissues to the antibody drugs, which carries the risk of various types of infusion-related toxic reactions and syndromes, including death [84]. ADCs such as T-DM1 and T-DXd commonly cause severe, life-threatening, or even fatal drug-related adverse events, especially hematologic and gastrointestinal toxic effects such as neutropenia, anemia, and fatigue, with almost all treated patients experiencing treatment-associated adverse events of any grade [75,85,86,87]. In contrast, intratumoral delivery of bsAb mRNA-LNPs limits the exposure primarily within the tumors while achieving high and durable local antibody expression, and largely avoids the nonspecific exposure-associated adverse events. In fact, intratumoral delivery allows immediate access of the bsAb mRNA-LNPs to lymphoid structures within the tumors, which enhances the activation of antitumor immune responses [88]. In addition, the HER2-CD3-Fc bsAb activates and recruits T cells to the tumors and harnesses T cells’ potent cytotoxic activity for the targeted killing of the HER2-positive tumor cells, which not only enhances the durability of treatment efficacy, but also reduces the off-target toxicity [71,73]. In our ovarian cancer xenograft model, the intratumoral injection of HER2-CD3-Fc mRNA-LNPs triggered a potent antitumor effect and completely eliminated the HER2-positive A1847 tumors without showing any signs of toxic effects. Therefore, it is of great value and significance to evaluate the efficacy of the HER2-CD3-Fc mRNA-LNPs for the treatment of HER2-positive cancer in clinical trials.

## 5. Conclusions

In this study, we developed a novel HER2-CD3-Fc mRNA-LNP-based immunotherapy for HER2-positive cancers. The data show that the HER2-CD3-Fc mRNA-LNPs mediate robust and sustained expression of HER2-CD3-Fc bsAb in various types of target cells and induce highly potent T cell cytotoxicity and cytokine secretion against multiple HER2-positive tumor cells in vitro. The HER2-CD3-Fc bsAb-mediated antitumor effect is highly specific and strictly dependent on its binding to the HER2 epitope, as tumor cells lacking HER2 become resistant to the bsAb-induced T cell cytotoxicity. The bsAb also requires binding to CD3 for T-cell recruitment, and the ablation of CD3 binding abrogates the bsAb’s ability to elicit antitumor activity. Intratumoral injection of the HER2-CD3-Fc mRNA-LNPs triggers a strong antitumor response and completely blocks HER2-positive A1847 tumor growth without any signs of toxicity in an ovarian cancer xenograft model. These results suggest that this novel HER2-targeted therapy has the potential to effectively and safely treat HER2-positive cancer in the clinic.

## 6. Patents

The antibody sequences are included in the patent application.

## Figures and Tables

**Figure 2 vaccines-12-00808-f002:**
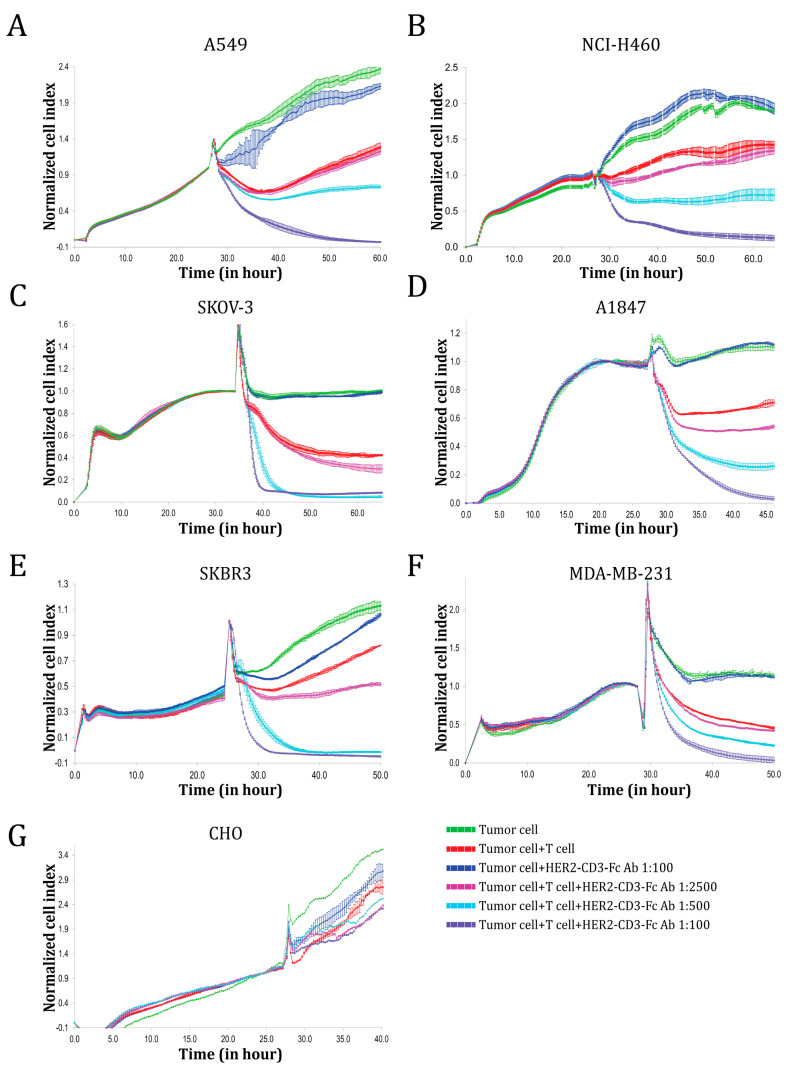
Functional analysis of the HER2-CD3-Fc bsAb using real-time cell analysis (RTCA). Target cells A549 (**A**), NCI-H460 (**B**), SKOV-3 (**C**), A1847 (**D**), SKBR3 (**E**), MDA-MB-231 (**F**), or CHO (**G**) were seeded in triplicate in a 96-well E-plate overnight. The next day, T cells and bsAb-containing HEK293S cell medium were added to the target cells. The ratio of the T cells to the target cells was 10:1. The impedance of the target cell monolayer was monitored with the RTCA system. Mean ± SD impedance is plotted and normalized to the time of T cell addition. Note the dose-dependent killing of the target cells by the T cells in the presence of the HER2-CD3-Fc bsAb. The high concentration of the bsAb induced almost complete lysis of the target tumor cells.

**Figure 3 vaccines-12-00808-f003:**
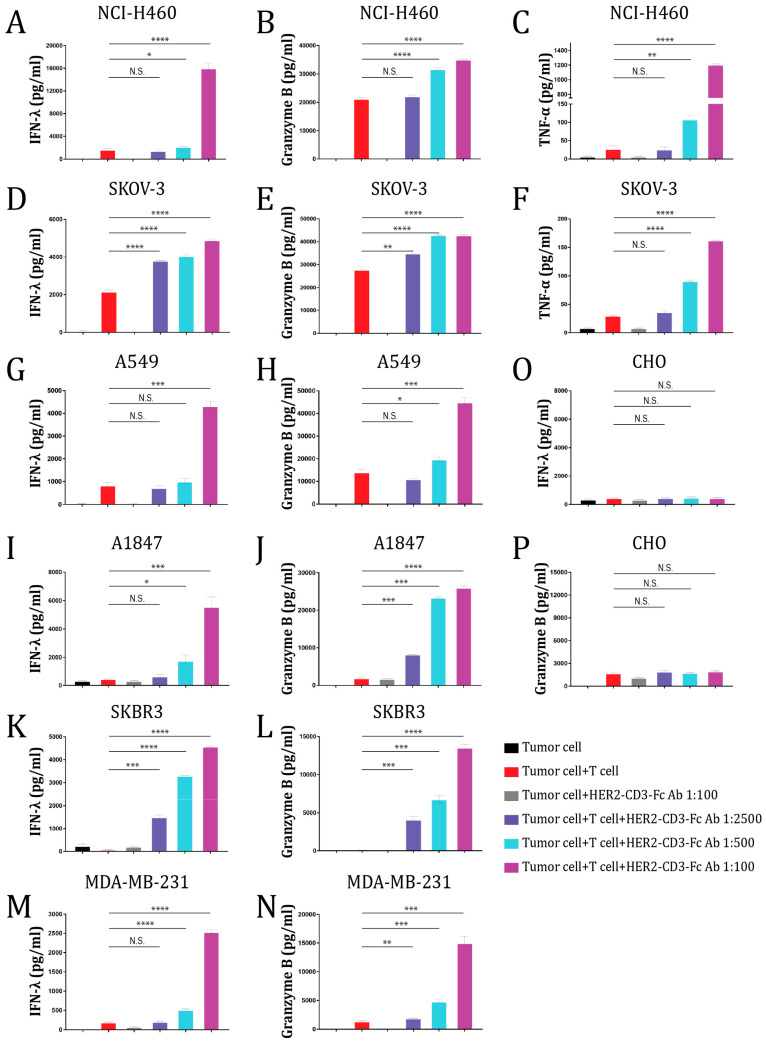
The HER2-CD3-Fc bsAb induces T cells to secrete effector molecules and cytokines against HER2-positive target cells. A total of 24 h after the addition of the T cells and bsAb-containing HEK293S medium to the target cells NCI-H460 (**A**–**C**), SKOV-3 (**D**–**F**), A549 (**G**,**H**), A1847 (**I**,**J**), SKBR3 (**K**,**L**), MDA-MB-231 (**M**,**N**), or CHO (**O**,**P**), the culture medium was collected and analyzed using ELISA for IFN-λ, granzyme B, and TNF-α levels. Data are representative of at least two independent experiments (three replicates per group). Data are presented as the means ± SD. A *p* value < 0.05 was considered statistically significant (* *p* < 0.05, ** *p* < 0.01, *** *p* < 0.001, and **** *p* < 0.0001). A *p* value ≥ 0.05 was statistically not significant (N.S.).

**Figure 4 vaccines-12-00808-f004:**
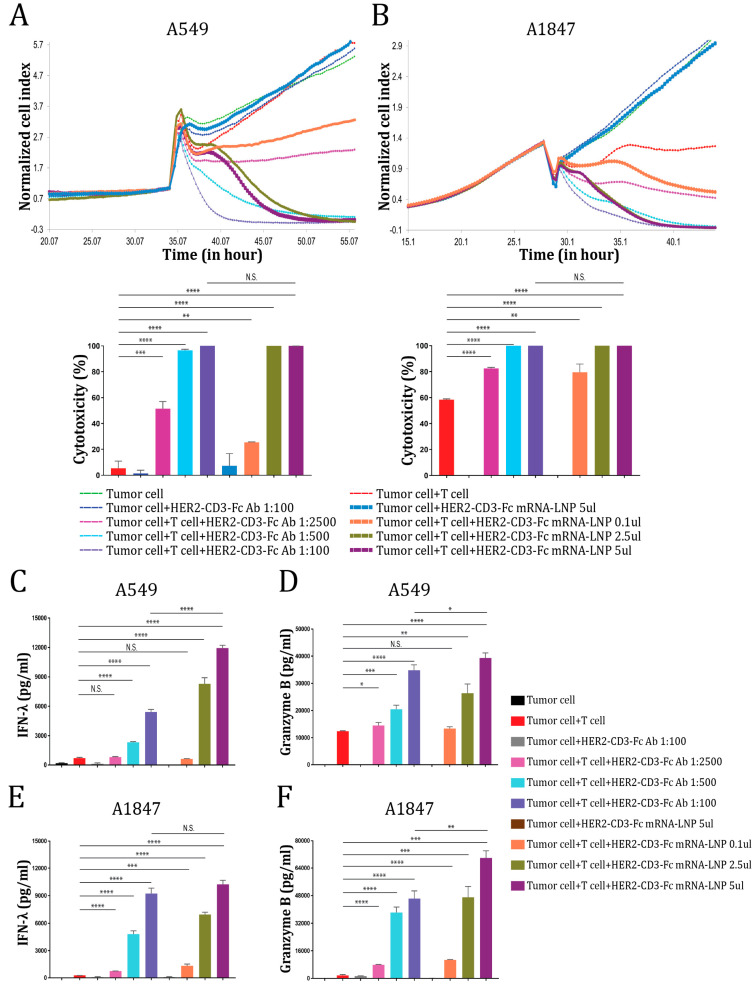
HER2-CD3-Fc mRNA-LNPs induce T cell cytotoxicity against HER2-positive tumor cells. (**A**,**B**) The target cells A549 (**A**) or A1847 (**B**) were seeded in triplicate in a 96-well E-plate overnight. The next day, T cells and either bsAb-containing HEK293S cell medium or bsAb mRNA-LNPs were added to the target cells. The ratio of the T cells to the target tumor cells was 10:1. The impedance of the target cell monolayer was monitored with the RTCA system. Mean ± SD impedance is plotted and normalized to the time of T cell addition. Cytotoxicity calculated based on the cell indexes of tumor cells alone and tumor cells with various treatments at the end of the recording is shown in the lower panel. (**C**,**F**) The culture medium from A (**C**,**D**) and B (**E**,**F**) were analyzed using ELISA for IFN-λ (**C**,**E**) and granzyme B (**D**,**F**) levels. Data are representative of at least two independent experiments (three replicates per group). Data are presented as the means ± SD. A *p* value < 0.05 was considered statistically significant (* *p* < 0.05, ** *p* < 0.01, *** *p* < 0.001, and **** *p* < 0.0001). A *p* value ≥ 0.05 was statistically not significant (N.S.).

**Figure 5 vaccines-12-00808-f005:**
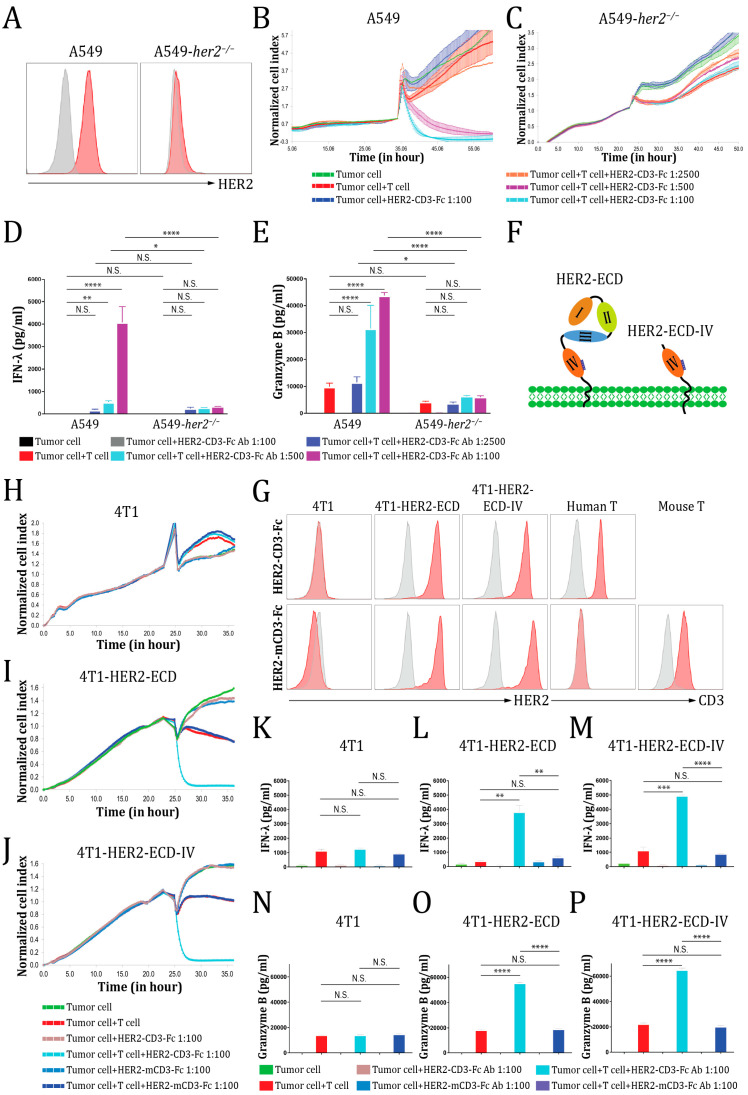
The HER2-CD3-Fc bsAb mediates a highly specific antitumor effect dependent on HER2. (**A**,**G**) Binding analysis of the HER2-CD3-Fc bsAb. The A549 *her2* knockout cells (**A**) were generated using the CRISPR/Cas9 system, while the 4T1-HER2-ECD and 4T1-HER2-ECD-IV cells (**G**) were generated by transduction of 4T1 cells with lentiviruses expressing the extracellular domain and transmembrane region of human HER2 (HER2-ECD, aa 1–675) or a truncated HER2-ECD with the epitope-containing subdomain IV, the transmembrane region, and a short intracellular tail (HER2-ECD-IV, aa 509–700), respectively. The HER2 expression on these cells and CD3 expression on the human or mouse T cells were verified by flow cytometry using 1:10-diluted bsAb-containing HEK293S cell medium (red histograms) or control HEK293S medium (gray histograms), followed by APC anti-human Fc secondary Ab. The HER2-mCD3-Fc bsAb was also used to stain the 4T1 tumor cells and T cells. (**B**–**E**,**H**–**P**) Analysis of the HER2-CD3-Fc bsAb-mediated T cell cytotoxicity. The target tumor cells A549 (**B**), A549-*her2*^−/−^ (**C**), 4T1 (**H**), 4T1-HER2-ECD (**I**), and 4T1-HER2-ECD-IV (**J**) were seeded in triplicate in a 96-well E-plate overnight. The next day, T cells and indicated bsAb-containing HEK293S cell medium were added to the tumor cells. The ratio of the T cells to the tumor cells was 10:1. The impedance of the target cell monolayer was monitored with the RTCA system. Mean ± SD impedance is plotted and normalized to the time of T cell addition. The culture medium was collected and analyzed using ELISA for IFN-λ (**D**,**K**–**M**) and granzyme B (**E**,**N**–**P**) levels. (**F**) Schematic presentation of the HER2-ECD and the HER2-ECD-IV. The HER2 extracellular domain consists of four tandem subdomains (I–IV). Subdomain IV contains the epitope (denoted by a blue bar) recognized by the HER2-CD3-Fc bsAb. The green dots represent the cell membrane phospholipid bilayer. Data in (**B**–**E**,**H**–**P**) are representative of at least two independent experiments (three replicates per group). Data in (**D**,**E**,**K**–**P**) are presented as the means ± SD. A *p* value < 0.05 was considered statistically significant (* *p* < 0.05, ** *p* < 0.01, *** *p* < 0.001, and **** *p* < 0.0001). A *p* value ≥ 0.05 was statistically not significant (N.S.).

**Figure 6 vaccines-12-00808-f006:**
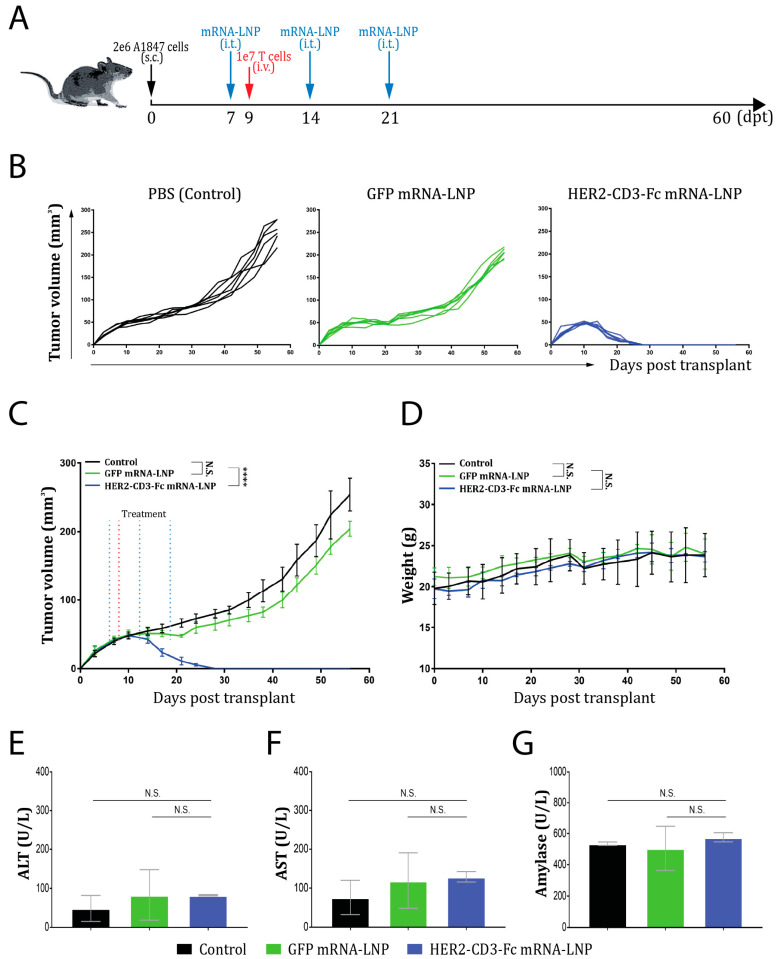
Intratumoral injection of HER2-CD3-Fc mRNA-LNPs induces a strong antitumor response in an NSG mouse ovarian cancer xenograft model. (**A**–**G**) A total of 2 × 10^6^ A1847 cells were inoculated into NSG mice subcutaneously (s.c.) on day 0 of the study, then 20 µL of PBS, GFP mRNA-LNPs (containing 1 µg mRNA), or HER2-CD3-Fc mRNA-LNPs (containing 1 µg mRNA) were injected intratumorally (i.t.) on days 7, 14, and 21. On day 9, 1 × 10^7^ human T cells were injected intravenously (i.v.). The tumor sizes were measured with calipers and used to calculate tumor volume with the formula (*width*^2^ × *length*)/2. The tumor volumes (**B**,**C**) and mouse body weights (**D**) were monitored and recorded every three days. Blood samples were collected at the end of the study and tested for toxicology biomarkers alanine aminotransferase (ALT) (**E**), aspartate aminotransferase (AST) (**F**), and amylase (**G**). Each line in (**B**) represents a tumor. Data are presented as the means ± SD in (**C**,**D**). N.S., not significant; ****, *p* < 0.0001.

## Data Availability

The datasets used and/or analyzed during the current study are available from the corresponding authors upon reasonable request.

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
