# Peer review of "HER2-CD3-Fc Bispecific Antibody-Encoding mRNA Delivered by Lipid Nanoparticles Suppresses HER2-Positive Tumor Growth"

_vaccines, 2024, doi:10.3390/vaccines12070808_

Round 1

Reviewer 1 Report

Comments and Suggestions for Authors

The authors present the anti-cancer efficacy in vitro and in vivo (xenogenic mouse model) of an mRNA-encoded bispecific antibody. The synthetic mRNA is formulated in LNPs. For the in vivo study, it is injected intratumoral. The in vitro work is very extensive and clearly presented. On the contrary, the in vivo results are limited to one mouse model and injection route. The major comment is: it is not clear why the mRNA in LNP was injected intra-tumor and not intra-venous. Could the author comment on the injection route? Why did they chose the cumbersome intratumor injection route? Did they try intravenous?

Minor comments:

The title is not clear. It could sound like LNPs are targeted by a bispecific. How about "HER2-CD3-Fc bispecific antibody-coding mRNA delivered by lipid nanoparticles suppresses HER2-positive tumor growth"?

This type of work (synthetic mRNA coding for an anticancer bispecific antibody) has been published seven years ago. It should be mentioned in the introduction and listed in the references: Stadler et al Elimination of large tumors in mice by mRNA-encoded bispecific antibodies. Nature Medicine, 12 Jun 2017, 23(7):815-817

Could the author give as supplementary information the exact protein sequence (or nucleic acid sequence) of the bispecific?

The authors wrote "targeted delivery" (for example lines 568 and 571). The delivery by LNP is not at all targeted. It is on the contrary totally unspecific. It can be assumed that several cell types take up the LNP and express the mRNA. Could the authors give arguments that they "target" or alternatively remove any mention of targeting?

Lines 590-591 what allows the author to say that intratumoral delivery limits exposure primarily within the tumors? Do they have in vivo imaging data with mRNA coding firefly luciferase in LNP injected intratumoral and showing indeed that exposure to mRNA is primarily in the tumors?

Author Response

Comment 1: The authors present the anti-cancer efficacy in vitro and in vivo (xenogenic mouse model) of an mRNA-encoded bispecific antibody. The synthetic mRNA is formulated in LNPs. For the in vivo study, it is injected intratumoral. The in vitro work is very extensive and clearly presented. On the contrary, the in vivo results are limited to one mouse model and injection route. The major comment is: it is not clear why the mRNA in LNP was injected intra-tumor and not intra-venous. Could the author comment on the injection route? Why did they chose the cumbersome intratumor injection route? Did they try intravenous?

Response 1: We would like to thank the reviewer for valuable comments. We briefly addressed the advantages of intratumoral injection in the original manuscript and added a little bit more discussion and few extra references in the Discussion section on this topic in the revised manuscript. Intravenous injection would cause expose the mRNA-LNPs to the blood cells and many other tissues, leading to expression of the HER2 bsAb in systematic circulation and many other tissues such as liver; this can lead to its rapid degradation and potentially undesired toxic reactions. In comparison, intratumoral injection is more desirable due to the local enrichment of the mRNA-LNPs and the bsAb, which can achieve better therapeutic effects with much less amount of mRNA-LNPs. Intratumoral injection can also limit exposure of the mRNA-LNPs and the bsAb by other tissues, and thus reduce the potential toxicity. We added the review in the discussion by Blanco et al., Int J Mol Sci. 2023 Feb; 24(3): 2676 which discussed the advantages of intratumoral injection of antibody drugs for solid tumors. 

Comment 2: The title is not clear. It could sound like LNPs are targeted by a bispecific. How about "HER2-CD3-Fc bispecific antibody-coding mRNA delivered by lipid nanoparticles suppresses HER2-positive tumor growth"?

Response 2: We agree with the reviewer and revised the title according to the suggestion.

Comment 3: This type of work (synthetic mRNA coding for an anticancer bispecific antibody) has been published seven years ago. It should be mentioned in the introduction and listed in the references: Stadler et al Elimination of large tumors in mice by mRNA-encoded bispecific antibodies. Nature Medicine, 12 Jun 2017, 23(7):815-817 

Response 3: We would like to thank the reviewer for the valuable suggestion. We have included the reference (Stadler et al, Nature Medicine, 2017) in the introduction of revised manuscript.  

Comment 4: Could the author give as supplementary information the exact protein sequence (or nucleic acid sequence) of the bispecific?

Response 4: Thank you for the comment. We have included the exact amino acid sequence of the bispecific antibody in the supplementary Figure S1.

Comment 5: The authors wrote "targeted delivery" (for example lines 568 and 571). The delivery by LNP is not at all targeted. It is on the contrary totally unspecific. It can be assumed that several cell types take up the LNP and express the mRNA. Could the authors give arguments that they "target" or alternatively remove any mention of targeting?

Response 5: We would like to thank the reviewer for the comment. We intended to mean “targeted delivery of the mRNA-LNP to tumors through intratumoral injection”. We have changed the wording “targeting” to avoid the ambiguity in the revised manuscript. 

Comment 6: Lines 590-591 what allows the author to say that intratumoral delivery limits exposure primarily within the tumors? Do they have in vivo imaging data with mRNA coding firefly luciferase in LNP injected intratumoral and showing indeed that exposure to mRNA is primarily in the tumors?

Response 6: We would like to thank the reviewer for the valuable suggestion. We mean that intratumoral delivery of the mRNA-LNPs primarily limits exposure of the mRNA-LNPs within the tumors, which differs from the intravenous injection that results in systematic exposure of the mRNA-LNPs to various organs and tissues. The imaging with firefly luciferase is a great idea to visually observe the tumor-restricted exposure after intratumoral injection of firefly luciferase mRNA-LNPs. We will test this in our next mouse study.

Reviewer 2 Report

Comments and Suggestions for Authors

Excellent manuscript. I cannot recommend any revisions. 

Author Response

Comment 1: Excellent manuscript. I cannot recommend any revisions. 

Response 1: Thank you very much for your comment. 

Reviewer 3 Report

Comments and Suggestions for Authors

This paper designed a mRNA construct of a bispecific antibody against HER2 and CD3, formed the mRNA-LNP nanoparticle, and evaluated the expression and cytotoxicity against HER2 positive cells. Overall the technology is novel in the mRNA-LNP field, the data is relatively sound to demonstrate the conclusion. I would make a few revisions before the acceptance:

1. Usually in the mRNA-LNP field, modified nucleotide was used to avoid triggering strong immune reaction. I would suggest include another group of mRNA with modified nucleotide as a control group to justify the cytokine production is indeed from the ADCC rather than the non-modified nucleotide.

2. it would be better if the author investigated how the T cell polarizes (CD4 or CD8 expansion) after mixed with the HER2 positive cells and the bispecific antibody.

3. I think in the overall experimental design, more control group is needed. For example, for the bispecific antibody, I would argue that we need a control group of HER2 antibody, CD3 antibody, bispecific antibody with or without the Fc region to prove that all the part is important for the designed function.

4. Figure2 and Figure 4AB is really hard to read, I would suggest extract the important data from the time curve and make a bar graph to make it easier to interpret

Author Response

Comment 1: Usually in the mRNA-LNP field, modified nucleotide was used to avoid triggering strong immune reaction. I would suggest include another group of mRNA with modified nucleotide as a control group to justify the cytokine production is indeed from the ADCC rather than the non-modified nucleotide.

Response 1: We would like to thank the reviewer for the valuable suggestion. We used modified methylpseudo-UTP during mRNA in vitro transcription to decrease immune reaction to exogenous mRNA. We included additional data in Figure 5 to provide more control results—we added some data about a control bsAb HER2-mCD3-Fc where we replaced the human CD3 scFv of our bsAb with a mouse version of CD3 scFv. We found that the HER2-mCD3-Fc bound to both 4T1-HER2-ECD and 4T1-HER2-ECD-IV cells as well as to mouse T cells, but not to HER-2-negative 4T1 and human T cells; however, the HER2-mCD3-Fc failed to direct human T cells to target 4T1-HER2-ECD and 4T1-HER2-ECD-IV cells and secrete cytokines IFN-λ and granzyme B. Additionally, we showed in Figures 5 and S4 that loss of her2 almost completely abrogated the bsAb-induced T cell cytotoxicity and cytokine production against the A549 and A1847 tumor cells. These results indicate that the bsAb-mediated T cell cytotoxicity and cytokine production is dependent on the antibody-antigen interaction and the ADCC effect, but not the mRNA. 

Comment 2: it would be better if the author investigated how the T cell polarizes (CD4 or CD8 expansion) after mixed with the HER2 positive cells and the bispecific antibody.

Response 2: We would like to thank the reviewer for the valuable comment. This is a good suggestion. We will analyze CD8 and CD4 polarization in depth in both the in vitro and in vivo assays in the next study. 

Comment 3: I think in the overall experimental design, more control group is needed. For example, for the bispecific antibody, I would argue that we need a control group of HER2 antibody, CD3 antibody, bispecific antibody with or without the Fc region to prove that all the part is important for the designed function.

Response 3: We would like to thank reviewer for the comment. Our results show that in the absence of T cells, the bsAb had no obvious effect on tumor growth, indicating that HER2 or CD3 antibody alone had no cytotoxic effect on tumor cells. In addition, we revised Figure 5 and added additional control results. We replaced human CD3 scFv with a mouse CD3 scFv and generated the control bsAb HER2-mCD3-Fc, which losted its binding to human T cells (Figure 5G, lower panel). We added additional curves and bars for Figure 5H-Q to show CD3-dependent and human T cell-dependent cytotoxicity against the tumor cells, indicating indispensable role of the human CD3 scFv for the proper functioning of the bsAb. In addition, the bsAb-mediated HER2-dependent antitumor effect shown in Figures 5 and S4 also demonstrates a necessary role of the HER2 scFv for the bsAb-mediated antitumor activity. The Fc region is well known for its role in maintaining antibody stability and mediating effector functions (PMIDs: 34932587, 29375551). Our previous study (PMID: 37345198) demonstrated that the Fc region does not affect the EpCAM-CD3-Fc bsAb-mediated T cell-dependent antitumor activity in vitro. But given its pivotal roles in antibody stability and effector function, the Fc region is expected to be critical for enhanced antibody drug therapeutic efficacy in vivo.

Comment 4: Figure2 and Figure 4AB is really hard to read, I would suggest extract the important data from the time curve and make a bar graph to make it easier to interpret

Response 4: We would like to thank the reviewer for the helpful comment. We added the bar graphs for Figure 4A-B as suggested. We also included additional explanation for Figure 2 in its legend on RTCA result to make it easier to understand. 

We are thankful to the reviewer as these comments greatly helped improve our revised manuscript!